# Psychological Characteristics and Addiction Propensity According to Content Type of Smartphone Use

**DOI:** 10.3390/ijerph17072292

**Published:** 2020-03-29

**Authors:** Jinhee Lee, Joung-Sook Ahn, Seongho Min, Min-Hyuk Kim

**Affiliations:** 1Department of Psychiatry, Yonsei University Wonju College of Medicine, 20 Ilsan-ro, Gangwon-do, Wonju 26426, Korea; jinh.lee95@yonsei.ac.kr (J.L.); mchorock@yonsei.ac.kr (S.M.); 2Division of Child and Adolescent Psychiatry, Yonsei University Wonju College of Medicine, 20 Ilsan-ro, Gangwon-do, Wonju 26426, Korea

**Keywords:** suicide, suicide attempts, intervention, case management, smartphone use, addiction

## Abstract

The aim of this study was to evaluate the association between content type of smartphone use and psychological characteristics and addiction propensity, including the average time of smartphone use and problematic smartphone use. Data were obtained from the 2017 Korea Youth Risk Behavior Web-Based Survey, a nationally representative survey of middle- and high-school students (*n* = 62,276). The content type of smartphone use was divided into four categories: (1) Study, (2) Social-Networking Services (SNS), (3) Game, and (4) Entertainment. The association of depressive mood and suicidal ideation with content type of smartphone use was analyzed, using multiple and binary logistic regression analyses, respectively. The relationship between content type of smartphone use and time spent on smartphone use and problematic smartphone use was analyzed by using multiple logistic regression, adjusted for related covariables. The results of this study revealed that depressive mood and suicidal ideation were significantly associated with the SNS smartphone use group, compared with the other groups. Our results also indicate that the SNS group showed higher addiction propensity, such as overuse and experiencing adverse consequences of smartphone use.

## 1. Introduction

Smartphones are perceived as indispensable information and communication tools in daily life for many people and are now the most frequently used technology worldwide [1]. For adolescents, in particular, who are sensitive to new technology and media use, smartphones have become an important part of their life. Recent studies estimate that 84% of adolescents in Japan [2] and 97% of adolescents in Switzerland [3] have their own smartphone. Furthermore, similar to substance or other types of behavioral addictions, adolescents are known to be vulnerable to smartphone addiction. A prior study reported that 60% of adolescents in the UK are highly addicted to their smartphones [2], and the rate of smartphone addiction among adolescents was double for adults in South Korea [2]. Prior studies have suggested some neurobiological evidences of the vulnerability toward smartphone addiction among adolescents, such as the dual processing model and an imbalance between the go and stop networks [4,5].

Smartphones are useful for multiple purposes, including study, information searching, social communication, and entertainment [6]. Compared with the traditional forms of computer and internet use, the portability and connectivity of smartphones give users easier access to information and entertainment content—nearly anytime and anywhere. These characteristics can also make people more vulnerable to behavioral addiction [7] in the form of habitual checking or excessive use of smartphones. Previous studies have reported that excessive smartphone use in adolescents is associated with psychopathologies (i.e., depression, anxiety, high-stress levels, and low mood) and behavioral problems [8,9], because adolescents are easily affected by external stimulus, interpersonal issues, and emotional changes. Another study on young adults suggested that excessive smartphone use is related to high stress, and it is also inversely related to academic performance, as well as life satisfaction [10].

Previous studies have used the smartphone addiction scale, smartphone usage time, or the frequency of use to clarify the relationship between addictive smartphone use and adverse effects on physical and mental health. However, despite the possible associations between the purpose of smartphone use and the risk of addictive behaviors, little is known regarding the relative impact of the content type of smartphone use on addiction and adverse consequences of smartphone use, including adolescent mental health. Andreassen and her colleagues have suggested that addictive online behaviors, including both addictive social networking and video gaming, are associated with underlying psychiatric disorders, such as ADHD, OCD, anxiety, and depression [11]; however, the differences in specific associations according to the purpose of its use have not been clarified. Another study has indicated that social-networking addiction and internet-gaming disorder can augment the symptoms of each other and simultaneously contribute to deterioration of overall psychological health in a similar fashion [12]. Some evidence shows that internet addiction comprises strongly directed internet activities, such as excessive online-video-game playing, excessive use of online pornography, or online shopping, and there are indeed different forms of internet addiction [13,14,15].

Thus, we aim to investigate the association between content type of smartphone use and adolescent mental health, with the hypothesis that psychological characteristics and addiction propensity are related with content type of smartphone use. This study examines the average time of smartphone use and problematic smartphone use, using a school-based, nationally representative dataset of the Korean adolescent population.

## 2. Materials and Methods

### 2.1. Methods

#### Study Population and Source of Data

Data on the study population were obtained from the 13th Korea Youth Risk Behavior Web-Based Survey (KYRBS), which was administered in 2017 by the Korean Ministry of Education, Science and Technology; the Ministry of Health and Welfare; and the Korea Centers for Disease Control and Prevention. KYRBS is a self-reported anonymous online survey of a nationally representative sample of Korean adolescents (aged 12–18 years) [16]. The sample design of this survey used a stratified multistage cluster strategy with 123 questions divided into 15 sections inquiring about health-related behaviors and mental and physical health. In the 13^th^ KYRBS, 64,991 students from 800 middle and high schools were randomly selected, and 62,276 (31,636 boys and 30,640 girls) students (95.8% response rate) from 799 schools responded to the survey [17]. Participants were provided with identification numbers and were guaranteed anonymity, and written informed consent was obtained from each participant after the survey had been fully explained. All data used in this study were fully anonymized before we accessed them. This consent procedure was approved by the Institutional Review Board of the Korea Centers for Disease Control and Prevention (2014-06EXP-02-P-A).

### 2.2. Measures

#### 2.2.1. Content Type of Smartphone Use

The exposure variable, content type of smartphone use, was assessed by the question “In the last 30 days, please select only one service that you used mainly, when using your smartphone”, and the answers were classified into four categories: (1) Study; (2) Social-Networking Services (SNS) (e.g., messaging and chat, communities, and social networks); (3) Game; and (4) Entertainment (e.g., watching movies, reading comics and fiction, listening to music, creating User-Created Content and videos).

#### 2.2.2. Sociodemographic and General Characteristics

The sociodemographic characteristics reported included age, sex, residential area, and family economic status of the participant. Respondents who lived in the country or rural areas were categorized as “Rural”; those who lived in small, middle-sized or large cities were categorized as “Urban”. Family economic status was assessed by the question “What is your family economic status?” The five possible response categories, very high/high/middle/low/very low, were grouped into three categories, for the purpose of our analysis: high (very high or high), middle (middle), and low (very low or low) [18]. Sleep hours were divided into two categories: under 6 h; and 6 or more hours. Physical activity was divided into two categories, Yes/No, from the question “In the last 7 days, did you have physical activity with higher heart rate than usual?”

#### 2.2.3. Psychological Characteristics

Subjective stress was measured by the question “How much stress do you usually feel?” The five possible response categories of very high//high/middle/low/very low were grouped into three categories: high (very high or high), middle (middle), and low (very low or low). Current alcohol consumption was assessed by the question “How many days during the past 30 days did you drink more than one cup of alcohol? (None/1–2 days/3–5 days/6–9 days/10–19 days/20–29 days/Every day)” Respondents who responded “None” were classified as not current alcohol drinkers, and those who responded between “1–2 days” and “Every day” were classified as current alcohol drinkers. The current cigarette smoking was assessed by using the following question: “How many days during the past 30 days did you smoke a cigarette? (None/1–2 days/3–5 days/6–9 days/10–19 days/20–29 days/Every day)”. Respondents who responded “No” to the question were classified as not current cigarette smokers, and those who responded between “1–2 days” and “Every day” were classified as current cigarette smokers.

Depressed mood among the subjects was assessed by the question “In the past year, have you ever felt so sad or despaired that your feelings disturbed everyday life for two whole weeks?” Subjects responded with the following: (1) “No, I never felt it” or (2) “Yes, I have felt it”. We also examined whether the subjects had suicidal ideations with the question “In the past year, did you ever seriously consider attempting suicide?” Subjects responded with the following: (1) “No, I never thought of it” or (2) “Yes, I have thought of it”.

#### 2.2.4. Addiction-Propensity-Related Factors

The average time spent using a smartphone was assessed by the question “On an average school day, how many hours do you use a smartphone?” According to the results of the previous study [3], the use of a smartphone over 5 h a day was defined as “smartphone overuse”. Participants were also asked the following: “In the last 30 days, have you experienced severe conflicts with family due to your smartphone usage?” and “In the last 30 days, have you experienced severe conflicts with friends due to your smartphone usage?”, (Yes/No), which would suggest a tolerance that is one of the important factors of smartphone addiction. They were also asked whether they had experienced poor academic performance due to smartphone use (Yes/No), by the question “In the last 30 days, were there any difficulties in your academic performance due to your smartphone usage?”, which would suggest a daily disturbance due to smartphone addiction [1].

#### 2.2.5. Statistical Analyses

The participants’ general characteristics according to each content type of smartphone use were summarized by using either a one-way analysis of variance for continuous variables or a chi-squared test with Bonferroni correction for categorical variables. The relationships of the four different groups with psychological factors and problematic smartphone use were analyzed by using Pearson’s chi-square. Subsequently, a multiple logistic regression analysis was performed to identify the associations between content type of smartphone use with depressed mood, suicidal ideation, and overuse of smartphones. General characteristics that showed a significant difference in the chi-square test were mutually adjusted for the analysis. Two-tailed analyses were conducted, and *p*-values lower than 0.05 were considered significant. Adjusted odds ratios (AORs) and 95% confidence intervals (CIs) were calculated. All statistical analyses were performed by using SPSS software (version 23.0, IBM Corp., Armonk, NY, USA).

## 3. Results

The descriptive characteristics according to the content type of smartphone use are presented in Table 1. Results of chi-square analysis revealed that there were significant differences depending on age, sex, residential area, family economic status, sleep hours, and physical activity by content type of smartphone use. The “Study” group was more likely to be older, live in large cities, and have a higher family economic status. The “SNS” group had a higher prevalence of female respondents and a lower prevalence of physical activity. The “Game” group was more likely to be younger, boys, living in rural areas, sleeping less than 6 h, and less physically active. The “Entertainment” group had a higher prevalence of low family economic status compared to other groups (all *p* < 0.001).

The psychological characteristics of participants according to content type of smartphone use are presented in Table 2. The “SNS” group had a higher prevalence of high subjective stress level, current cigarette smoking, and current alcohol drinking. The “SNS” group also had significantly higher prevalence of depressive mood and suicidal ideation compared to other groups. The “Game” group had the lowest proportion of depressive mood and suicidal ideation among groups (all *p* < 0.001).

The average amount of time spent using a smartphone was greater in the “SNS” group (322.17 ± 228.90 min/day) than in other groups and lower in the “Study” group (176.97 ± 173.06 min/day). The proportion of adverse consequences of smartphone use, including conflicts with family, conflicts with friends, and poor academic performance due to smartphone use, were higher in the “SNS” group (59.2%, 27.6%, and 58.4% respectively), whereas the “Study” group had a lower prevalence of adverse consequences (41.8%, 20.1%, and 43.0% respectively) (Table 3).

Compared to the “Study” group, the “SNS” group was significantly more likely to report a depressive mood (AOR 1.36; 95% CI 1.24–1.49) and suicidal ideation (AOR 1.49; 95% CI 1.32–1.69). The “Entertainment” group also showed a positive association with suicidal ideation (AOR 1.20; 95% CI 1.06–1.35), and the “Game” group showed a negative association with depressive mood (AOR 0.77; 95% CI 0.69–0.85) (Table 4).

The AORs for smartphone overuse (> 5 h per day) were 4.57 (95% CI, 4.20–4.98), 2.24 (95% CI, 2.05–2.45) and 2.60 (95% CI, 2.40–2.81) in the “SNS” group, “Game” group, and “Entertainment” group, respectively (Table 5).

## 4. Discussion

This study examined the association of psychological characteristics and addiction propensity with content type of smartphone use, within a relatively large convenience sample of adolescents in Korea. The results of this study revealed that depressive mood and suicidal ideation are significantly associated with higher SNS use, compared with smartphone use for games, study, and entertainment. Our results also suggested that the “SNS” group showed higher addiction propensity, including overuse and adverse consequences of smartphone use.

The results of this study expanded upon and shared similarities with previous findings on the relationship between mental health and SNS use. A prior systematic review by Frost et al. reported associations between SNS use (i.e., Facebook) and mental-health outcomes, such as alcohol use, addiction, anxiety, and depression [19]. Several studies have indicated that the prolonged use of SNS may be related to signs and symptoms of depression, and some authors have indicated that certain SNS activities might be associated with low self-esteem, especially in children and adolescents [20,21,22,23]. On the other hand, our results were contrary to a previous study that reported that the use of non-social smartphone features (i.e., news consumption, entertainment, and relaxation) were most related to depression and problematic smartphone use [24]. Furthermore, a prior study by our research group reported a potential protective effect from moderate use (1–2 h) of smartphones for social purposes (i.e., SNS and messaging) in regard to suicide attempts [1]. In these contexts, we should also consider the positive psychological effects of SNS use. In this study, we did not simply divide content type of smartphone use as social and non-social, but instead we compared detailed non-social uses: study, game, entertainment, and SNS. According to our results, content type of smartphone use should not be classified simply as social and non-social use but should also take into account the detailed characteristics of SNS use and various other tasks, including differences in the effects of mental health on adolescents.

There has been wide discussion on the potential causes for depressive mood resulting from increased time on SNS. The most commonly used mediator to explain the association between SNS use and depression is self-esteem. It is an important factor in developing and maintaining mental health and overall quality of life, and low self-esteem is associated with numerous mental illnesses, including depression and addiction [25,26]. Some authors have presented that individuals higher in narcissism and lower in self-esteem also showed more online activity, including self-promotional content such as SNS [27]. On the other hand, there is the that hypothesis feelings of depression can be predicted indirectly by SNS addiction [21]. Authors have indicated that SNS allows the user to get virtual community gratification and gain gratification from creating a self-image online. Based on the uses and gratifications theory, SNS use can lead to SNS addiction, as the functions available to the users allow them to gain instant gratification from using the service, which in turn could lead to excessive use.

Contrary to previous research that indicated a negative association between online-game use and adolescent mental health [28,29], the current study did not find that smartphone-game use was associated with depressive mood and suicidal ideation. The results might reflect the characteristics of categorization and reference group of study. The “Game” group of this study included those who enjoyed “gaming” more than other contents of the smartphone, but it does not mean that they had a “gaming addiction”. Specifically, if a person performs gaming in a regular pattern, the person may relieve his/her stress. However, if a person overly performs gaming, he or she may have increased psychological problems, as shown in the literature. On the other hand, because of the statistically low number of “non-smartphone users”, we used “Study” as a reference group. Studying does not mean the person cannot be addicted to it, and using “Study” as the reference can create some biases. For example, a person who is over-studying may have increased distress. Therefore, we cannot capture whether gaming is related to increased distress if studying is associated with high distress.

Furthermore, smartphone-game use predicted problematic smartphone use compared to the “Study” group, but showed a weak association compared to the “SNS” or “Entertainment” groups. Smartphone games are somewhat different from computer-based online games, allowing users to access them anywhere, anytime, but there is a limit to the use of tools for the games. There have been a number of studies on problematic game use, and recently, a WHO ICD-11 proposal for a new category named “Gaming Disorder” [30]. However, most of the studies so far have been limited to computer-based online games [31,32,33]. Furthermore, considering the recent trend that the use of entertainment, such as the use of YouTube, is particularly popular among adolescents and has become dominant in the media market worldwide [34,35], the results of the current study indicated the necessity for further studies about game and entertainment on the smartphone. Smartphones, which are relatively simple tools compared to conventional computers, may be better suited for simple functions, such as watching videos, than for more complex tasks, such as playing games, which may result in adolescents indulging in media instead.

The present study has a number of limitations that should be considered when interpreting the findings. First, due to the cross-sectional nature of national surveys, the present findings have limitations in explaining the causal inferences between content type of smartphone use and psychological characteristics. Further studies with sufficient time for investigation are needed to develop a clear understanding about the association of psychological characteristics and addiction propensity with content type of smartphone use. Second, the psychological characteristics and internet use were measured through the ad hoc questions rather than mental-health experts’ assessment or validated scales, because the data were collected through the participants’ self-reports, and therefore, some reporting bias could have occurred. Moreover, because the group was divided only for one main purpose of smartphone use, we could not distinguish those who performed two or more content types. Third, in our study, the addictive propensity was estimated only by time spent using a smartphone, not by the scales for smartphone addiction. In addition, most variables in the study, including conflicts with family/friends, and poor academic performance due to smartphone use, were surveyed on the basis of a self-reported questionnaire, which has inherent limitations regarding the validity of the data and the recall bias. However, in the previous study, excessive smartphone use was validated as the most powerful independent predictor of smartphone addiction [8], and we can use this to estimate the propensity to addiction. Fourth, our data lacked information regarding the familiarity or personological profile of the participants that might affect individuals with mood disorder and/or addiction. Despite the limitations of this cross-sectional survey, the present study has some strengths. We used a multilevel multinomial logistic modeling approach based on a nationally representative sample of Korean adolescents, who have the highest smartphone ownership rate in the world. Moreover, the response rate to the survey was very high. To the best of our knowledge, this study is the first to report on the association of psychological characteristics and addiction propensity with the content type of smartphone use in adolescents.

## 5. Conclusions

In the present nationally representative sample of Korean adolescents aged 12–18 years, the significant and specific association between content type of smartphone use and the prevalence of depressed mood, suicidal ideation, and addiction propensity has been confirmed. Our findings indicate that SNS use was not only a stronger predictor of smartphone addiction than other content types, but also had stronger associations with negative psychological characteristics, including depressive mood and suicidal ideation. The results of this study suggest that careful consideration should be given to improving screening for the risk of smartphone addiction and mental health problems in adolescents, with a focus on SNS use. Further research is needed to understand the longitudinal impact of specific content type of smartphone use on the addictive behavior of adolescents, allowing for the development of effective prevention strategies and to help strengthen the current smartphone-use guidelines.

## Figures and Tables

**Table 1 ijerph-17-02292-t001:** General characteristics of participants, according to content type of smartphone use.

General Characteristics		Study (*n* = 4202, 10.7%)	SNS (*n* = 10,192, 25.9%)	Game (*n* = 7282, 18.5%)	Entertainment (*n* = 17,642, 44.9%)	*p*
Age (years)	Mean (SD)	15.58 ± 1.78	14.92 ± 1.66	14.31 ± 1.71	15.07 ± 1.76	< 0.001
Sex (%)						< 0.001
	Boys	62.9	28.9	87.9	50.8	
	Girls	37.1	71.1	12.1	49.2	
Region (%)						< 0.001
	Rural Area	7.3	7.2	8.5	7.8	
	Small City	45.6	47.3	50.3	47.3	
	Large City	47.1	45.5	41.2	44.9	
Family Economic Status (%)						< 0.001
	High	47.2	39.0	40.4	37.0	
	Middle	40.3	46.8	46.2	47.1	
	Low	12.5	14.2	13.4	15.9	
Sleep Hours (%)						< 0.001
	<6hrs	33.2	38.6	62.4	42.9	
	≥6hrs	66.8	61.4	37.6	57.1	
Physical Activity (%)						< 0.001
	No	34.2	38.9	32.0	36.5	
	Yes	65.8	61.1	68.0	64.0	

SNS: Social-Networking Services.

**Table 2 ijerph-17-02292-t002:** Psychological characteristics of participants, according to content type of smartphone use.

Psychological Characteristics		Study (*n* = 4202, 10.7%)	SNS (*n* = 10,192, 25.9%)	Game (*n* = 7282, 18.5%)	Entertainment (*n* = 17,642, 44.9%)	*p*
Subjective Stress						< 0.001
	High	37.1	41.1	30.3	38.0	
	Middle	40.9	42.0	43.2	42.1	
	Low	22.0	16.6	26.5	19.8	
Current Cigarette Smoking						0.694
	No	93.4	93.2	93.6	93.3	
	Yes	6.6	6.8	6.4	6.7	
Current Alcohol Drinking						< 0.001
	No	77.7	74.7	77.4	75.5	
	Yes	22.3	25.3	22.6	24.5	
Depressive Mood						< 0.001
	No	76.2	71.1	82.5	76.2	
	Yes	23.8	28.9	17.5	23.8	
Suicidal Ideation						< 0.001
	No	89.5	85.3	90.6	87.7	
	Yes	10.5	14.7	9.4	12.3	

SNS: Social-Networking Services.

**Table 3 ijerph-17-02292-t003:** Adverse consequences of smartphone use, according to content type of smartphone use.

Adverse Consequences		Study (*n* = 4202, 10.7%)	SNS (*n* = 10,192, 25.9%)	Game (*n* = 7282, 18.5%)	Entertainment (*n* = 17,642, 44.9%)	*p*
Average Spent Time Using a Smartphone (min/day)						< 0.001
	Mean (SD)	176.97 ± 173.06	322.17 ± 228.90	243.87 ± 191.73	255.10 ± 187.00	
Conflict with Family Members due to Smartphone Use (%)						< 0.001
	No	58.2	40.8	41.5	45.8	
	Yes	41.8	59.2	58.5	54.2	
Conflict with Friends due to Smartphone Use (%)						< 0.001
	No	79.9	72.4	74.1	78.4	
	Yes	20.1	27.6	25.9	21.6	
Poor Academic Performance due to Smartphone Use (%)						< 0.001
	No	57.0	41.6	51.9	47.4	
	Yes	43.0	58.4	48.1	52.6	

SNS: Social-Networking Services.

**Table 4 ijerph-17-02292-t004:** Multivariable logistic regression analysis of content type of smartphone use, for depressive mood and suicidal ideation.

Content Type	Depressive Mood		Suicidal Ideation	
	Crude OR	Adjusted OR *	Crude OR	Adjusted OR *
	OR (95% CI)	OR (95% CI)	OR (95% CI)	OR (95% CI)
Study	Ref.	Ref.	Ref.	Ref.
SNS	1.30 (1.20–1.41)	1.36 (1.24–1.49)	1.46 (1.31–1.64)	1.49 (1.32–1.69)
Game	0.68 (0.62–0.74)	0.77 (0.69–0.85)	0.87 (0.77–0.99)	0.95 (0.82–1.09)
Entertainment	1.00 (0.92–1.08)	1.02 (0.94–1.12)	1.19 (1.07–1.32)	1.20 (1.06–1.35)

* Adjustment for age, sex, region of residence, family economic status, sleep hours, and physical activity; SNS: Social-Network Services, OR: odds ratio.

**Table 5 ijerph-17-02292-t005:** Multivariable logistic regression analysis of content type of smartphone use, for smartphone overuse (more than 5 h per day).

Content Type	Crude OR	Adjusted OR
	OR (95% CI)	OR (95% CI)
Study	Ref.	Ref.
SNS	4.67 (4.32–5.05)	4.57 (4.20–4.98)
Game	2.22 (2.04–2.40)	2.24 (2.05–2.45)
Entertainment	2.68 (2.49–2.88)	2.60 (2.40–2.81)

* Adjustment for age, sex, region of residence, family economic status, sleep hours, and physical activity; SNS: Social-Networking Services, OR: odds ratio

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
