# Peer review of "Psychological Characteristics and Addiction Propensity According to Content Type of Smartphone Use"

_ijerph, 2020, doi:10.3390/ijerph17072292_

Round 1

Reviewer 1 Report

The topic is timely and interesting. The manuscript is well written, but among the limitations I would consider the lack of information regarding the familiality of the involved individuals for mood disorders and/or addiction. In addition, we do not have data regarding the personological profile of the respondents (particularly important when self-rating aspects such as perceived level of stress). 

Author Response

The topic is timely and interesting. The manuscript is well written, but among the limitations I would consider the lack of information regarding the familiality of the involved individuals for mood disorders and/or addiction. In addition, we do not have data regarding the personological profile of the respondents (particularly important when self-rating aspects such as perceived level of stress). 

Response: Thank you for your valuable opinion. We fully agree with the fact that our data has a lack of information regarding the familiarity or personological profile of the participants that might affect individuals for mood disorder and/or addiction. As recommended, we revised manuscript and added the limitation in the revised manuscript (Discussion lines 262-264): 

"Fourth, our data lacked information regarding the familiarity or personological profile of the participants that might affect individuals with mood disorder and/or addiction."

Reviewer 2 Report

This is an interesting paper focusing on an important topic. I think it can be considered for publication after minor revisions, as follows:

- In the introduction section, the authors state that “Smartphones have become indispensable in daily life”. I would rather write that they are “perceived” as indispensable.

- The authors correctly state that adolescents are at higher risk for smartphone addiction. However, they do not state that there are neurobiological issues underpinning this risk. I think the authors must include this specification using related references.

- I suggest including specific aims and hypotheses, which should be then commented in the discussion section.

- If I understand well, the psychological characteristics and internet use were measured through ad hoc questions. Why did the study choose this method instead of widely used (validated) measures? Please clarify. I understand that these measures were chosen by the original study whose raw data have now been used by the authors for this contribution, however, I think the reader should have some details on the rationale for the tools choosing.

- the reference list is appropriate but I think it could be richer and should include at least some major contributors in this field, such as, for example, Mark Griffiths and his collaborators (as visible here: https://www.ncbi.nlm.nih.gov/pubmed/31344851; https://www.ncbi.nlm.nih.gov/pubmed/24001297; https://www.ncbi.nlm.nih.gov/pubmed/31496849; https://www.ncbi.nlm.nih.gov/pubmed/29462976)

Author Response

1. In the introduction section, the authors state that “Smartphones have become indispensable in daily life”. I would rather write that they are “perceived” as indispensable.

Response: Thank you for indicating the confusion of our manuscript. To clarify this, we revised the sentencesas suggested in the revised manuscript (Introduction, lines 25-26):

"Smartphones are perceived as indispensable information and communication tools in daily life for many people and are now the most frequently used technology worldwide[1]."

2. The authors correctly state that adolescents are at higher risk for smartphone addiction. However, they do not state that there are neurobiological issues underpinning this risk. I think the authors must include this specification using related references.

Response: Thank you for your opinion with which we strongly agree. We added the sentences and revised the manuscript as suggested in the revised manuscript (Introduction, lines 33-35):

"Prior studies have suggested some neurobiological evidences of the vulnerability toward smartphone addiction among adolescents, such as the dual processing model and an imbalance between the go and stop networks[4,5]"

3. I suggest including specific aims and hypotheses, which should be then commented in the discussion section.

Response: We absolutely agree with your opinion that additional clarification is needed regarding the hypothesis and aim of the study. To clarify this, we added the following sentece in the revised the manuscript (Introduction, lines 62-65):

"Thus, we aim to investigate the association between content type of smartphone use and adolescent mental health, with the hypothesis that psychological characteristics and addiction propensity are related with content type of smartphone use."

4. If I understand well, the psychological characteristics and internet use were measured through ad hoc questions. Why did the study choose this method instead of widely used (validated) measures? Please clarify. I understand that these measures were chosen by the original study whose raw data have now been used by the authors for this contribution, however, I think the reader should have some details on the rationale for the tools choosing.

Response: We understand and agree with your opinion that our data has the limitation of using the ad hoc questions rather than the used measures. We added these limitations of the data more clearly in the limatation section. (Discussion, lines 229-231):

"Second, the psychological characteristics and internet use were measured through the ad hoc questions rather than mental health experts’ assessment or validated scales because the data were collected through the participants’ self-reports, and therefore, some reporting bias could have occurred."

5. The reference list is appropriate but I think it could be richer and should include at least some major contributors in this field, such as, for example, Mark Griffiths and his collaborators (as visible here: https://www.ncbi.nlm.nih.gov/pubmed/31344851; https://www.ncbi.nlm.nih.gov/pubmed/24001297; https://www.ncbi.nlm.nih.gov/pubmed/31496849; https://www.ncbi.nlm.nih.gov/pubmed/29462976)

Response: As recommnnded, we added the suggested references in the revised manuscript as follows.

[9] Cerniglia, L.; Griffiths, M.D.; Cimino, S.; De Palo, V.; Monacis, L.; Sinatra, M.; Tambelli, R. A latent profile approach for the study of internet gaming disorder, social media addiction, and psychopathology in a normative sample of adolescents. Psychology research and behavior management 2019, 12, 651.

[18] Cerniglia, L.; Guicciardi, M.; Sinatra, M.; Monacis, L.; Simonelli, A.; Cimino, S. The use of digital technologies, impulsivity and psychopathological symptoms in adolescence. Behavioral Sciences 2019, 9, 82.

[21] J Kuss, D.; D Griffiths, M.; Karila, L.; Billieux, J. Internet addiction: A systematic review of epidemiological research for the last decade. Current pharmaceutical design 2014, 20, 4026-4052.

Reviewer 3 Report

I read the manuscript entitled "Psychological characteristics and addiction propensity according to content type of smartphone use" with great interests as the topic is timely and interesting. The major strengths of this study include the use of a nationally representative sample with a large sample size and the relevant topic. I partially agree that current literature has rarely compared how different content types in the smartphone perform differently with psychological characteristics. Moreover, some methodological concerns should be clearly described in the manuscript. Therefore, the authors are recommended revising their manuscript according to my following comments.

  1. The authors should notice that some studies have examined how different types of behavioral addiction related to psychological characteristics. For example, Andreassen et al. (2016), Pontes (2017) have examined the relationships between psychological characteristics and two types of behavioral addiction (gaming and social media use). Although the authors have argued in the Discussion section that “gaming” could be performed by a computer instead of smartphone, I believe that prior research does not clearly distinguish the desktop game and the smartphone game. Therefore, some of the respondents in the previous studies could perform gaming on a smartphone. In this sense, the authors still need to reveal the information to the readers regarding the comparisons of the different types of behavioral addiction in the Introduction.

Ref:

Andreassen, C.S.; Billieux, J.; Griffiths, M.D.; Kuss, D.J.; Demetrovics, Z.; Mazzoni, E.; Pallesen, S. The relationship between addictive use of social media and video games and symptoms of psychiatric disorders: A large-scale cross-sectional study. Psychol. Addict. Behav. 2016, 30, 252–262.

Pontes, H.M. Investigating the differential effects of social networking site addiction and Internet gaming disorder on psychological health. J. Behav. Addict. 2017, 6, 601–610.

  1. I recommend the authors comparing the different content types of smartphone use with the concept of “specific internet addiction” as these different content types are “specific”. I understand that in the current study, the content type of studying is positive and may not be a common problem in “addiction”. However, we cannot rule out any possibility that a person is addicted to studying just like there are workaholic people. The authors may consider reviewing the following references to strengthen their Introduction.

Sha, P., Sariyska, R., Riedl, R., Lachmann, B., & Montag, C. (2018). Linking Internet Communication and Smartphone Use Disorder by taking a closer look at the Facebook and WhatsApp applications. Addictive Behaviors Reports, 100148.

Brand, M., Young, K. S., Laier, C., Wölfling, K., & Potenza, M. N. (2016). Integrating psychological and neurobiological considerations regarding the development and maintenance of specific Internet-use disorders: An Interaction of Person-Affect-Cognition-Execution (I-PACE) model. Neuroscience & Biobehavioral Reviews, 71, 252-266.

Brand, M., Young, K. S., & Laier, C. (2014). Prefrontal control and internet addiction: A theoretical model and review of neuropsychological and neuroimaging findings. Frontiers in Human Neuroscience, 8, 1–13. doi:10.3389/fnhum.2014.00375

Davis, R. A. (2001). A cognitive-behavioral model of pathological Internet use. Computers in Human Behavior, 17(2), 187–195. doi:10.1016/S0747-5632(00)00041-8

Montag, C., Bey, K., Sha, P., Li, M., Chen, Y.-F., Liu, W.-Y., … Reuter, M. (2015). Is it meaningful to distinguish between generalized and specific Internet addiction? Evidence from a cross-cultural study from Germany, Sweden, Taiwan and China. Asia-Pacific Psychiatry, 7(1), 20–26. doi:10.1111/appy.12122

  1. I did not see any reason why the authors wanted to use 5 hours to define smartphone overuse. There is also no such operational definition for the smartphone overuse in the Methods section. However, Results sections directly pop out the results related to the more than 5 hours of smartphone use.
  2. In the Discussion, ll204-218, the authors proposed one reason for their findings that smartphone gaming was not associated with depressive mood and suicidal ideation is the media in gaming (i.e., smartphone and computer). However, I think that authors may need to add another two possible reasons: (1) “gaming addiction” and “gaming” are different. Specifically, if a person performs gaming in regular pattern, the person may relieve his/her stress. However, if a person overly performs gaming, he or she may have increased psychological problems as shown in the literature. (2) As I mentioned earlier, studying does not mean it cannot be addicted to. Therefore, using “studying” as the reference may be biased. For example, a person who is over studying may have increased distress. Therefore, we cannot capture whether gaming is related to increased distress if studying is associated with high distress. The best gold standard (i.e., reference group) in the logistic regression models should be “did not use smartphone”; then, we can understand whether the use of the content is related to distress.
  3. Another two limitations should be mentioned. First, the authors cannot distinguish those who have performed two or more content types. It is very likely that people using social network paly smartphone gaming a lot. Similarly, those using social network may watch a lot of movies or clips. It is somehow hard to distinguish between these content types. Second, this is a cross-sectional study; therefore, the common problem is the weak evidence in causal relationship.

Author Response

1. The authors should notice that some studies have examined how different types of behavioral addiction related to psychological characteristics. For example, Andreassen et al. (2016), Pontes (2017) have examined the relationships between psychological characteristics and two types of behavioral addiction (gaming and social media use). Although the authors have argued in the Discussion section that “gaming” could be performed by a computer instead of smartphone, I believe that prior research does not clearly distinguish the desktop game and the smartphone game. Therefore, some of the respondents in the previous studies could perform gaming on a smartphone. In this sense, the authors still need to reveal the information to the readers regarding the comparisons of the different types of behavioral addiction in the Introduction.

Response: Thank you for your kind suggestion and we absolutely agree with your opinion. As recommended we added the sentences using the suggested references as follows. (Introduction, lines 52-58)

"Andreassen C. S. and his colleagues have suggested that addictive online behaviors including both addictive social networking and video gaming are associated with underlying psychiatric disorders, such as ADHD, OCD, anxiety, and depression[11]; however, the differences in specific associations according to the purpose of its use have not been clarified. Another study has indicated that social networking addiction and internet gaming disorder can augment the symptoms of each other and simultaneously contribute to deterioration of overall psychological health in a similar fashion[12]." 

2. I recommend the authors comparing the different content types of smartphone use with the concept of “specific internet addiction” as these different content types are “specific”. I understand that in the current study, the content type of studying is positive and may not be a common problem in “addiction”. However, we cannot rule out any possibility that a person is addicted to studying just like there are workaholic people. The authors may consider reviewing the following references to strengthen their Introduction.

Response: Thank you for indicating the confusion of our manuscript. To clarify this, we added the sentences and revised the manuscript as suggested in the revised manuscript (Introduction, lines 58-61):

"Some evidences show that internet addiction comprises strongly directed internet activities such as excessive online video game playing, excessive use of online pornography, or online shopping, and there are indeed different forms of internet addiction[13,14]."

3. I did not see any reason why the authors wanted to use 5 hours to define smartphone overuse. There is also no such operational definition for the smartphone overuse in the Methods section. However, Results sections directly pop out the results related to the more than 5 hours of smartphone use.

Response: As recommended, we added the definition of smartphone overuse in the Methods section as follows (Method, lines 123-124).

"According to the results of the previous study[18], the use of a smartphone over 5 hours a day was defined as the “smartphone overuse."

4. In the Discussion, ll204-218, the authors proposed one reason for their findings that smartphone gaming was not associated with depressive mood and suicidal ideation is the media in gaming (i.e., smartphone and computer). However, I think that authors may need to add another two possible reasons: (1) “gaming addiction” and “gaming” are different. Specifically, if a person performs gaming in regular pattern, the person may relieve his/her stress. However, if a person overly performs gaming, he or she may have increased psychological problems as shown in the literature. (2) As I mentioned earlier, studying does not mean it cannot be addicted to. Therefore, using “studying” as the reference may be biased. For example, a person who is over studying may have increased distress. Therefore, we cannot capture whether gaming is related to increased distress if studying is associated with high distress. The best gold standard (i.e., reference group) in the logistic regression models should be “did not use smartphone”; then, we can understand whether the use of the content is related to distress.

Response: We absolutely understand and agree with your opinion that we need some more refined interpretation of the results. We thank you for your kindly letting us amend it. As recommended, we added the possible reasons as follows : (Discussion, lines 223-232)

"The results might reflect the characteristics of categorization and reference group of study. The “game group” of this study included those who enjoyed “gaming” more than other contents of smartphone, but it does not mean that they had “gaming addiction.” Specifically, if a person performs gaming in regular pattern, the person may relieve his/her stress. However, if a person overly performs gaming, he or she may have increased psychological problems as shown in the literature. On the other hand, because of the statistically low number of “non-smartphone user,” we used “studying” as a reference group. Studying does not mean it cannot be addicted to, and using “studying” as the reference can make some bias. For example, a person who is over studying may have increased distress. Therefore, we cannot capture whether gaming is related to increased distress if studying is associated with high distress."

5. Another two limitations should be mentioned. First, the authors cannot distinguish those who have performed two or more content types. It is very likely that people using social network paly smartphone gaming a lot. Similarly, those using social network may watch a lot of movies or clips. It is somehow hard to distinguish between these content types. Second, this is a cross-sectional study; therefore, the common problem is the weak evidence in causal relationship.

Response: Thank you for indicating the limitations of our description. We absolutely agree with your opinion that it is hard the distinguish between the two or more content types that are performed at the same time. According to the reviewer’s comment, we added these limitations and revised the manuscript as follows (Discussion, lines 247-249; 254-256):

"First, due to the cross-sectional nature of national surveys, the present findings have limitations in explaining the causal inferences between content type of smartphone use and psychological characteristics. "

"Moreover, because the group was divided only for one main purpose of smartphone use, we could not distinguish those who performed two or more content types."

Round 2

Reviewer 3 Report

I appreciate the authors carefully revised their work. However, some minor points exist.

1. line 52. Andreassen is a woman not a man. Therefore, please correct "his colleagues" to "her colleagues".
2. line 58. It should be "some evidence" instead of "some evidences"
3. Please add one recently published paper in the IJERPH at line 61.
Wong, H.Y.; Mo, H.Y.; Potenza, M.N.; Chan, M.N.M.; Lau, W.M.; Chui, T.K.; Pakpour, A.H.; Lin, C.-Y. Relationships between Severity of Internet Gaming Disorder, Severity of Problematic Social Media Use, Sleep Quality and Psychological Distress. Int. J. Environ. Res. Public Health 202017, 1879.

Author Response

1. line 52. Andreassen is a woman not a man. Therefore, please correct "his colleagues" to "her colleagues"

Response: We corrected "his colleagues" to "her colleagues"

2. line 58. It should be "some evidence" instead of "some evidences"

Response: We corrected "some evidences" to "some evidence"

3. Please add one recently published paper in the IJERPH at line 61.
Wong, H.Y.; Mo, H.Y.; Potenza, M.N.; Chan, M.N.M.; Lau, W.M.; Chui, T.K.; Pakpour, A.H.; Lin, C.-Y. Relationships between Severity of Internet Gaming Disorder, Severity of Problematic Social Media Use, Sleep Quality and Psychological Distress. Int. J. Environ. Res. Public Health 202017, 1879

Response: We added the reference as suggested.
